# Low Light Stress Increases Chalkiness by Disturbing Starch Synthesis and Grain Filling of Rice

**DOI:** 10.3390/ijms23169153

**Published:** 2022-08-15

**Authors:** Qiuping Li, Fei Deng, Yuling Zeng, Bo Li, Chenyan He, Youyun Zhu, Xing Zhou, Zinuo Zhang, Li Wang, Youfeng Tao, Yu Zhang, Wei Zhou, Hong Cheng, Yong Chen, Xiaolong Lei, Wanjun Ren

**Affiliations:** 1State Key Laboratory of Crop Gene Exploration and Utilization in Southwest China, Chengdu 611130, China; 2Key Laboratory of Crop Eco-Physiology and Farming System in Southwest China, Ministry of Agriculture and Rural Affairs, Chengdu 611130, China; 3College of Agronomy, Sichuan Agricultural University, Chengdu 611130, China

**Keywords:** rice, chalkiness, starch synthesis, low light stress, grain filling

## Abstract

Low light stress increases the chalkiness of rice; however, this effect has not been fully characterized. In this study, we demonstrated that low light resulted in markedly decreased activity of ADP-glucose pyrophosphorylase in the grains and those of sucrose synthase and soluble starch synthase in the early period of grain filling. Furthermore, low light also resulted in decreased activities of granule-bound starch synthase and starch branching enzyme in the late period of grain filling. Therefore, the maximum and mean grain filling rates were reduced but the time to reach the maximum grain filling rates and effective grain filling period were increased by low light. Thus, it significantly decreased the grain weight at the maximum grain filling rate and grain weight and retarded the endosperm growth and development, leading to a loose arrangement of the amyloplasts and an increase in the chalkiness of the rice grains. Compared to the grains at the top panicle part, low light led to a greater decrease in the grain weight at the maximum grain filling rate and time to reach the grain weight at the maximum grain filling rate at the bottom panicle part, which contributed to an increase in chalkiness by increasing the rates of different chalky types at the bottom panicle part. In conclusion, low light disturbed starch synthesis in grains, thereby impeding the grain filling progress and increasing chalkiness, particularly for grains at the bottom panicle part.

## 1. Introduction

Rice grain quality has attracted considerable attention in recent years [1,2]. A high degree of rice grain translucency without a chalky appearance is a key indicator of rice quality, and better quality fetches a higher premium [3,4]. Chalkiness, which describes the opacity of rice grains, is measured using two indicators, chalky grain rate (CGR) and chalkiness degree (CD), which are closely associated with the milling, appearance, cooking, and consumption qualities of rice grains [5,6].

Chalkiness is primarily determined by the structure and arrangement of amyloplasts. Specifically, when amyloplasts are loosely arranged due to insufficient grain filling and delayed grain development, the light transmittance of the grains decreases; thus, they show a greater degree of chalkiness [7,8]. Chalkiness is a quantitative trait that is controlled by multiple genes [3], such as *Chalk5* [9], *OsPK2* [10], *qPCG1* [11], and *qACE9* [12]. An imbalance in the expression levels of these genes, which are related to starch biosynthesis, is also related to rice chalkiness [9,13]. For example, the suppression of *starch synthase I* (*SSI)* markedly increases amylose content and granule-bound starch synthase (GBSS) activity in rice grains, thereby significantly increasing the CGR and CD owing to the disturbance in the structure and arrangement of starch granules [14]. Furthermore, the downregulation of *starch branching enzyme IIb* (*SBEIIb*) gene expression results in the loose arrangement of round-shaped starch granules and the production of grains with a higher level of chalkiness or opacity [15].

Additionally, rice chalkiness is not only determined by the genetic background of the rice species but is also influenced by environmental factors [4,16,17], such as the temperature [18], water supply [19], and fertilizer application [20]. During the grain filling stage, high temperatures decrease the enzymatic activities of soluble starch synthase (SSS), sucrose synthase (SuS), and GBSS, thereby impeding grain filling and reducing starch accumulation in the endosperm, increasing the number of loosely arranged starch granules [14,21], thereby increasing the CGR and CD. Further, CGR and CD are significantly increased by drought stress [22]. Indeed, long-term water shortage accelerates leaf senescence during the grain filling stage, resulting in a decrease in grain growth potential and the grain filling rate and a consequent increase in chalkiness [19]. It has also been observed that chalkiness is strongly associated with the grain filling process.

Light, an essential environmental factor that affects plant morphogenesis, directly influences the photosynthetic rate of plant leaves and regulates the carbon metabolism of crops [23,24,25]. However, a decrease in surface solar radiation (low light [LL] stress) due to climate change and environmental pollution has become a challenge worldwide [24,26]. Thus, it has been observed that LL decreases leaf thickness by reducing the number of palisade layers and spongy parenchyma tissues, thereby reducing the photosynthetic rate of leaves [27,28]. This results in an imbalance in the accumulation and redistribution of photosynthetic products, which contributes to a significant decrease in rice grain yield [27,29,30]. Further, LL also substantially influences the development of rice grains by decreasing the activities of SSS and GBSS, leading to variations in the internal structure of the rice endosperm [5,14]. Therefore, LL increases both the CGR and CD of rice by inducing the production of more loosely packed amyloplasts with more airspaces owing to the poor uniformity of starch granules [5,8]. However, the impact of LL on the grain filling characteristics of rice, particularly its relationship to rice chalkiness, has not yet been extensively investigated.

Therefore, in this study, we conducted a field control light intensity study in Wenjiang, Sichuan, China from 2018 to 2021 with the following primary objectives: (a) to investigate starch synthesis and characteristics of the filling progress of grains under LL, (b) to investigate the impact of LL on the variations in grain chalkiness at different panicle parts, and (c) to evaluate the relationship of starch synthesis and grain filling with chalkiness in rice. Our results would provide insights for improving rice grain quality in low light intensity regions.

## 2. Results

### 2.1. Variation in Rice Grain Chalkiness under Low Light Stress

#### 2.1.1. Variation in Chalky Grain Rate and Chalkiness Degree under Low Light Stress

The panicle part significantly influenced the CGR between 2018 and 2020 and the CD in 2020 (Table 1). Compared with the CGR and CD corresponding to the grains at the top panicle part, those corresponding to the grains at the bottom panicle part increased by 50.40–141.14% and 0.60–71.32%, respectively. Furthermore, relative to the control plants, LL significantly increased the CGR by 96.40–259.71% and CD (except for the CD of the grains at the top panicle part in 2018) by 109.61–323.46%. Additionally, the effect of LL on the chalkiness varied with the panicle part. Compared with the grains at the top panicle part, LL resulted in a greater increase in the CGR and CD of the rice grains at the bottom panicle part.

#### 2.1.2. Variation in Chalkiness Types under Low Light Stress

The rates of different chalkiness types at different panicle parts in 2018 and 2019 are shown in Figure 1. The rice grains at the bottom panicle part showed higher rates of white-core, white-belly, white-back, mixed-white, and full-white chalkiness than those at the top panicle part. Meanwhile, in CK, white-core chalkiness was predominant at both panicle parts. However, LL significantly increased the white-core, white-belly, and white-back rates in 2018 and white-core and mixed-white rates in 2019 at the top panicle part. LL also significantly increased the mixed-white, white-core, and white-belly rates at the bottom panicle part in 2018 and 2019. Moreover, the rates of full-white and white-back chalkiness in 2019 also increased due to LL. These results indicated that LL had a stronger influence on the variation in the chalkiness type of the grains at the bottom panicle part than it did at the top panicle part. The increase in the rates of the different kinds of chalkiness, particularly the rates of mixed-white and full-white chalkiness, could be attributed to the increase in the CD of the rice grains. Moreover, the results of scanning electron microscopy of grains showed that LL markedly hindered the development of starch granules in rice, which resulted in an increase in the chalkiness of the rice due to an increase in the numbers of loosely arranged spherical amyloplasts with a spherical appearance in the grains (Figure 2). In addition, compared with the grains at the top panicle part, more irregular amyloplasts were observed in the grains at the bottom part of the panicle under LL, which contributed to the increase in their chalkiness.

### 2.2. Enzyme Activity Related to Starch Synthesis under Low Light Stress

The activities of the enzymes associated with starch synthesis initially increased and then decreased during grain filling (Figure 3). In general, the activities of these enzymes were markedly influenced by the light condition. LL markedly decreased the activity of ADP-glucose pyrophosphorylase (AGPase) at each sampling time point by 8.13–39.52%. Compared with the full sunlight (control) condition, the SuS and SSS enzyme activities showed delay in reaching their peaks under LL treatment by 6 and 3 days, respectively. LL also reduced the activities of SuS and SSS before 9 days after flowering (DAF) by 4.57–20.62% and 3.34–12.19%, respectively, but increased their activity after 9 DAF by 5.90–25.69% and 3.49–56.74%, respectively. Furthermore, LL resulted in an increase in (0.52–15.21%) and a decrease (6.32–19.86%) in GBSS activity before and after 9 DAF, respectively. Furthermore, the peak activity of SBE was reached 3 days in advance, with an increase in activity from 6 to 9 DAF but a decrease in activity after 12 DAF.

### 2.3. Rice Grain Filling Characteristics under Low Light Stress

#### 2.3.1. Grain Weight and Grain Filling Rate under Low Light Stress

The appearance of rice grains at 5 to 40 DAF is shown in Figure 4a. Compared with the control, LL markedly slowed down the growth and development of the rice grains, especially at the bottom panicle part. To analyze grain filling, the dry weight of the grain was fitted. The dynamic variation in rice grain weight with DAF fitted well with an S-shaped curve with *R*^2^ > 0.992 (Figure 4b,d and Table 2) in 2020 and 2021, indicating that the Richards’ growth equation could sufficiently describe the grain filling dynamics. Further, the grain weight and grain filling rate of the rice grains differed significantly with panicle part and light treatment. Compared with the grains at the bottom panicle part, those at the top panicle part showed greater weights and ultimate growth (A) values; this observation could be attributed to the higher grain filling rate at the top panicle part (Figure 4c). LL decreased the grain filling rate at both the panicle parts (Figure 4c) in 2020 and at the bottom panicle parts (Figure 4e) in 2021, thereby decreasing the grain weight and A by 5.18–14.39% and 4.89–15.28% at 40 DAF, respectively. Further, owing to the greater decrease in the grain filling rate at the bottom panicle part, LL resulted in a more significant decrease in the grain weight and A at the bottom panicle part than at the top panicle part.

#### 2.3.2. Grain Filling Characteristics of Rice under Low Light Stress

Panicle part (excluding I values) and light treatment (excluding active grain filling period values) significantly affected the grain filling characteristics in 2020, and light treatment significantly affected these parameters in 2021 (Table 3). Moreover, in 2020, the relationship between panicle part and light treatment significantly influenced the mean grain filling rates, maximum grain filling rates, grain weight at maximum grain filling rate, time for reaching the maximum filling rate, and active grain filling period, and, compared with the relative initial growth potential, mean grain filling rates, maximum grain filling rates, and grain weight at maximum grain filling rate of the grains at the top panicle part, the values of these parameters for those at the bottom panicle part were significantly decreased, while the I, time for reaching the maximum filling rate, active grain filling period, and effective grain filling period values increased. LL significantly increased the active grain filling period of the grains at the top panicle part in 2020 and significantly increased the active grain filling period of the grains at the bottom panicle part in 2021. Further, it also significantly increased the relative initial growth potential, time for reaching the maximum filling rate, and effective grain filling period values of the grains at both panicle parts in 2020 by 18.04–31.25%, 12.11–46.47%, and 11.24–15.38%, respectively. In 2021, the time for reaching the maximum filling rate and effective grain filling period values of the grains at the bottom panicle parts significantly increased by 12.39% and 12.60%, respectively. Conversely, there was a decrease in the mean grain filling rates, maximum grain filling rates, I, and grain weight at maximum grain filling rate across the panicle parts in 2020 by 9.93–12.28%, 11.21–12.28%, 8.27–8.67%, and 11.24–15.38%, respectively. Similarly, in 2021, these four indicators showed a decrease in the grains at the bottom panicle parts by 20.12%, 20.84%, 3.82%, and 16.70%, respectively. Furthermore, LL caused a greater variation in the relative initial growth potential, grain weight at maximum grain filling rate, and time for reaching the maximum filling rate of the grains at the bottom panicle part than for those at the top in 2020. This observation suggested that LL treatment significantly limited grain filling, particularly for grains at the bottom panicle part. Moreover, the results for the grains at the bottom panicle part in 2021 further verified again that LL stress limited grain filling.

#### 2.3.3. Grain Filling Characteristics at Different Stages

The grain filling process was divided into the early (0–t_1_), middle (t_1_–t_2_), and late (t_2_–t_3_) periods (Table 4). Panicle part and light treatment significantly affected the inflection points, as well as the number of days of grain filling in 2020. Further, the light treatment also significantly affected the mean grain filling rate and contribution rates of the different periods of the grain filling process in 2020. Additionally, light treatment significantly affected the inflection points, number of days, mean grain filling rate, and contribution rate of the different periods in 2021. Compared with the grains at the top panicle part, the inflection point values, days of the early, middle, and late periods, and contribution rate (except the contribution rate of the middle grain filling period) corresponding to the grains at the bottom panicle part were significantly greater, while the mean grain filling rate and contribution rate of the late grain filling period values were significantly lower. Furthermore, LL significantly increased the value of the inflection points, leading to a 10.70–21.52%, 13.06–13.50%, and 15.70–19.78% increase in t_1_, t_2_, and t_3_, respectively, compared with the values corresponding to the full light-treated plants in 2020, and, in 2021, the values increased by 12.55%, 12.29%, and 14.43%, respectively. However, under LL, the mean grain filling rate at early grain filling period, mean grain filling rate at middle grain filling period, and mean grain filling rate at late grain filling period corresponding to both panicle parts decreased in 2020 by 31.23% to 44.54%, 11.61% to 14.19%, and 14.45% to 16.21%, respectively. This contributed to an increase in the number of days of the early, middle, and late periods for grains from both the panicle parts in 2020 by 10.70% to 21.52%, 2.21% to 16.40%, and 21.36% to 35.88%, respectively, as well as an increase in the contribution rate of the middle and late periods by 5.25% to 6.06% and 19.91% to 22.15%, respectively. The grains at the bottom panicle part showed similar changes in 2021, with mean grain filling rate at early grain filling period, mean grain filling rate at middle grain filling period, and mean grain filling rate at late grain filling period decreasing by 30.47%, 20.83%, and 21.70%, respectively, and ultimately resulting in an increase in the contribution rate of the middle grain filling period and contribution rate of the late grain filling period by 2.20% and 8.29%, respectively.

### 2.4. Relationships between Chalkiness and Grain Filling Characteristics

Correlation analysis showed that the CGR and CD were significantly negatively associated with the mean grain filling rates, maximum grain filling rates, grain weight at maximum grain filling rate, and mean grain filling rate and significantly positively associated with the time for reaching the maximum filling rate, effective grain filling period, early grain filling period, late grain filling period, and the inflection point values (Figure 5). Further, the CGR and CD were negatively correlated with I and the contribution rate of the early grain filling period, but the correlation with CGR was not statistically significant. The CD significantly increased with the contribution rate of the middle grain filling period and contribution rate of the late grain filling period, while the CGR significantly improved with t_1_–t_2_. These observations suggest that, the lower the grain filling rate, the higher the possibility of chalkiness. Further, faster and more efficient grain filling led to a lower likelihood of chalkiness.

## 3. Discussion

LL, caused by solar dimming and industrial development, significantly decreases rice grain yield and quality [27,28,29], and, in particular, increased chalkiness negatively affects the market value of rice [31,32]. However, the variation in rice chalkiness under LL has not been thoroughly investigated. In this study, we estimated the impact of LL on the starch synthesis and grain filling characteristics of rice, which are strongly associated with the chalkiness of rice under LL (Figure 5). Our results can facilitate the breeding of shade-tolerant rice varieties, as well as the optimization of cultivation techniques to improve rice quality under LL.

Chalkiness, which describes the opacity of rice grains, is determined by the genetic background of the rice grains, as well as the sensitivity of the rice grains to environmental changes [33]. For example, high temperatures accelerate the grain filling process of rice, and this results in insufficient grain filling and an increase in the occurrence of chalky rice grains [21,34]. Furthermore, drought stress significantly decreases the photosynthetic rate and water potential of rice, which contributes to a significant increase in both CGR and CD [22]. As the primary energy source for green plants, light directly affects the photosynthetic rate of leaves and influences plant growth and development, thereby affecting crop yield and quality [23,24,28]. LL decreases plant leaf CO_2_ transport capacity by reducing the stomatal density and electron transfer rate of plants [35,36], resulting in a decrease in the photosynthetic rate owing to the attenuation of Calvin-cycle-related enzyme activity, as well as the capacity for CO_2_ fixation [37,38,39]. Thus, an insufficient supply of filling substances is observed owing to LL interfering with the accumulation and redistribution of photosynthate [27,40].

LL caused the grain filling to be blocked, leading to a decrease in sucrose, starch, and amylose contents in the grains. As a result, the brown rice rate, milled rice rate, and peak starch viscosity, cold glue viscosity, breakdown value, and glue consistency were significantly reduced under LL, while the protein content increased, eventually leading to poor rice quality [27,41]. This also contributed to an increase in the white-core and white-belly chalkiness rates of rice grains at both the top and bottom panicle part under LL, as well as the mixed-white, white-back, and full-white chalkiness rates at the bottom panicle part (Figure 1). Thereby, LL results in a significant increase in CGR and CD [5,8].

Rice endosperm is formed by flat cell groups arranged along the dorsal diameter of rice grains, which are rapidly filled with starch and storage proteins during grain filling [42]. Our results suggested that the grain filling characteristics of rice were closely associated with both the CGR and CD of rice (Figure 5), consistent with the results of Wei et al. [8,32]. In general, the rice grain filling process refers to starch biosynthesis and the accumulation of amyloplasts in the form of starch granules [5,43,44]. During the grain filling process, rice endosperm forms a well-developed flat cell population at 10 days after flowering, with abundant and neatly arranged amyloplasts [45], and the quantity and volume of the amyloplasts continue to increase throughout endosperm development [46]. The cells then crush each other to form an angular polyhedral structure with no interspaces, resulting in a transparent rice endosperm. However, poor grain filling, due to insufficient photosynthate supply, leads to an uneven enrichment and an irregular arrangement of the flat cell groups [5,27].

LL interferes with the photosynthate supply for grain filling [39,47], leading to the marked changes in the activities of five enzymes involved in regulating starch synthesis. The activity of AGPase, involved in energy metabolism, decreased under LL [48]. LL also decreased SuS activity at the early grain filling stage and caused a delay in reaching its peak activity, thus contributing to the lower energy supply for starch synthesis [48,49]. In addition, LL reduced the activities of GBSS and SBE after 12 DAF, resulting in a reduction in starch accumulation under LL [50,51]. Therefore, the mean grain filling rates and maximum grain filling rates were decreased and the time for reaching the maximum filling rate and effective grain filling period were increased, which contributed to a reduction in the starch granule uniformity of starch [5,8]. This results in a decrease in the number of amyloplasts and an increase in the number of round and loosely arranged amyloplasts (Figure 6), contributing to a considerable increase in the CGR and CD [11,16].

Moreover, the occurrence of chalkiness differed with respect to the panicle part. It is well known that the majority of grains with inferior quality were harvested from the bottom panicle part [44]. Compared with the grains at the top panicle part, those at the bottom panicle part showed significantly lower mean grain filling rates, maximum grain filling rates, and grain weight at maximum grain filling rate but greater time for reaching the maximum filling rate, active grain filling period, and effective grain filling period values. Thus, the grains from the bottom panicle part were characterized by a lower grain filling rate, poorer grain fullness, and lower grain weight [29,52]. Previous studies have indicated that LL has a greater effect on the formation of rice chalkiness at the bottom panicle part than at the top and middle panicle part [5,44]. Compared with the grains at the top panicle part, LL caused a greater decrease in the mean grain filling rates, maximum grain filling rates, and grain weight at maximum grain filling rate, and a greater increase in time for reaching the maximum filling rate, effective grain filling period, and CR of the grains at the bottom panicle part. This contributed to a further increase in both the CGR and CD of rice by increasing the rate and type of chalkiness for the grains at the bottom panicle part under LL.

## 4. Materials and Methods

### 4.1. Plant Material and Experimental Design

An LL field experiment was conducted in the Huihe farm of Sichuan Agricultural University (30°43′ N, 103°52′ E), Wenjiang, Sichuan, China from 2018 to 2021. The meteorological conditions (from sowing to harvest) of the experimental site were recorded from sowing to harvest (Figure 7). Further, the soil properties of the study area were determined as shown in Table 5. Huanghuazhan, a widely used and planting area conventional *indica* rice cultivar, was selected as the study material. To obtain a 53% LL condition, a layer of white cotton yarn screens was hung at approximately 0.5 m above the rice plants from flowering to maturity following the method described by Deng et al. [5,8]. The control plants were not shaded. The area of each subplot was 3 × 10 m. To conduct the experiments, 30-day-old seedlings were manually transferred on 23 May 2018, 23 May 2019, 21 May 2020, and 21 May 2021 at 33.3 × 20 cm spacing. Nitrogen (urea), basal (75.6 kg ha^−1^), and tillering (32.4 kg ha^−1^) fertilizers were applied. The initial fertilizer application rate at panicle initiation was 43.2 kg ha^−1^, and, at the booting stage, it was 28.8 kg ha^−1^. Further, at the basal stage, 90 kg ha^−1^ of P_2_O_5_ (superphosphate) and 90 kg ha^−1^ of K_2_O (potassium chloride) were applied, and 90 kg ha^−1^ of K_2_O was applied as a spikelet-promoting fertilizer. Chemical pesticides, such as tricyclazole, validamycin, avermectin, and penoxsulam, were used to prevent yield loss caused by diseases, insects, and weeds.

### 4.2. Labeling and Sampling

During flowering, 300 panicles with the same growth trend were labeled on the same day in each plot from 2018 to 2021. As previously described by Deng et al. [5], approximately 50 labeled panicles were sampled at maturity and the grains at the top and bottom panicle parts were separated to measure their chalkiness. Meanwhile, approximately 300 opening spikelets at the top and bottom parts of marked panicles were labeled for each plot using black and red paint pens in 2020, respectively. In 2021, approximately 600 opening spikelets at the bottom part of marked panicles were labeled for each plot.

### 4.3. Determination of Enzyme Activity Related to Starch Synthesis

The marked grains of approximately 30 labeled panicles were sampled on days 3, 6, 9, 12, 18, 24, and 30 after LL treatment in 2021. Each sample (0.1 g) was ground into a powder using liquid nitrogen. Thereafter, phosphate buffer saline (0.01 mol L^−1^; pH: 7.2–7.4) was added and mixed well, followed by centrifugation. The supernatant was collected, and enzyme-linked immunosorbent assay (ELISA) was performed using an ELISA kit (Meimian Technology Ins., Yancheng, China), as per the manufacturer’s instructions, to determine the activities of AGPase, SuS, SSS, GBSS, and SBE.

### 4.4. Measurement of Grain Filling Characteristics

In 2020, 10 labeled panicles were sampled on days 5, 10, 15, 20, 25, 30, 35, and 40, whereas, in 2021, they were sampled on days 6, 12, 18, 24, 30, 36, and 40. In each case, approximately 30 marked grains were randomly selected for the determination of the grain filling characteristics of the rice grains. Six grains were removed from the glumes at each sampling time and observed using a stereomicroscope (M165C, Leica, Wetzlar, Germany) in 2020. Furthermore, 20 grains from each panicle part in 2020 and from bottom part in 2021 were removed from the glumes and dried until a steady weight was obtained. Thereafter, Richards equation (Equation (1)) was used to measure the increase in the weight of grains at the top and bottom panicle parts, and relevant parameters were also calculated as previously described by Richards [53] and Wei et al. [44].
(1)$$W_{G}=A{\left(1+Be^{-Ct}\right)}^{-\frac{1}{D}}$$
where *W_G_* represents the dry weight of the grain (mg), *A* represents the maximum dry weight of the grain (mg), *t* represents the time after labeling (days), and *B*, *C*, and *D* represent equation parameters. The growth rate was derived from Equation (1) as follows: (2)$$Growth\;rate=\frac{ACBe^{-Ct}}{N{\left(1+Be^{-Ct}\right)}^{\frac{\left(D+1\right)}{D}}}$$

The respective equations (Equations (3)–(10)) were used to describe the filling characteristics of the rice grains as follows:(3)$$\mathrm{Relative}\;\mathrm{initial}\;\mathrm{growth}\;\mathrm{potential}=\frac{C}{D}$$
(4)$$\mathrm{Time}\;\mathrm{to}\;\mathrm{reach}\;\mathrm{the}\;\mathrm{maximum}\;\mathrm{filling}\;\mathrm{rate}\;(d)=\frac{lnB-lnD}{C}$$
(5)$$\mathrm{Mean}\;\mathrm{grain}\;\mathrm{filling}\;\mathrm{rate}\;(\mathrm{mg}\;{\mathrm{grain}}^{-1}\;d^{-1})=\frac{AC}{2\left(D+2\right)}$$
(6)$$\mathrm{Maximum}\;\mathrm{grain}\;\mathrm{filling}\;\mathrm{rate}\;(\mathrm{mg}\;{\mathrm{grain}}^{-1}\;d^{-1})=\frac{AC}{{\left(1+D\right)}^{\frac{D+1}{D}}}$$
(7)$$\mathrm{Grain}\;\mathrm{weight}\;\mathrm{at}\;\mathrm{maximum}\;\mathrm{grain}\;\mathrm{filling}\;\mathrm{rate}\;(\mathrm{mg})=A{\left(1+D\right)}^{-\frac{1}{D}}$$
(8)$$\mathrm{Grain}\;\mathrm{weight}\;\mathrm{at}\;\mathrm{maximum}\;\mathrm{grain}\;\mathrm{filling}\;\mathrm{rate}/\mathrm{maximum}\;\mathrm{grain}\;\mathrm{weight}\;(\%)=\frac{GW_{max}}{A}$$
(9)$$\mathrm{Active}\;\mathrm{grain}\;\mathrm{filling}\;\mathrm{period}\;(d)=\frac{2\left(D+2\right)}{C}$$
(10)$$\mathrm{Effective}\;\mathrm{grain}\;\mathrm{filling}\;\mathrm{period}\;(d)=\frac{lnB+4.595}{C}$$

The filling process could be divided into the early, middle, and late periods of grain filling [44]. Further, the growth rate equation (Equation (2)) had two inflection points that equated the second derivative related t to zero to yield the values of the two inflection points, t_1_ and t_2_ (Equations (11) and (12)). The grain filling process was also assumed to be completed when the grain weight reached 99% A, and this time point was designated t_3_, Equation (13).
(11)$$t_{1}\;=\;\frac{-ln\frac{D^{2}+3D+D\sqrt{D^{2}+6D+5}}{2B}}{C}$$
(12)$$t_{2}\;=\;\frac{-ln\frac{D^{2}+3D-D\sqrt{D^{2}+6D+5}}{2B}}{C}$$
(13)$$t_{3}\;=\;\frac{-ln(\frac{{\left(\frac{100}{99}\right)}^{D}-1}{B})}{C}$$

The grain filling stage was determined by three grain filling periods, calculated as: Early grain filling period (d) = t_1_ − 0 (14)
Middle grain filling period (d) = t_2_ − t_1_
(15)
Late grain filling period (d) = t_3_ − t_2_
(16)

Substituting t_1_, t_2_, and t_3_ into Equation (1), the corresponding grain weights W_1_, W_2_, and W_3_ were obtained, respectively. Additionally, the mean of the grain filling and contribution rates of the three stages were calculated as follows, respectively:(17)$$\mathrm{Mean}\;\mathrm{grain}\;\mathrm{filling}\;\mathrm{rate}\;\mathrm{at}\;\mathrm{early}\;\mathrm{grain}\;\mathrm{filling}\;\mathrm{period}\;(\mathrm{mg}\;{\mathrm{grain}}^{-1}\;d^{-1})\;=\frac{W_{1}}{t_{1}}$$
(18)$$\mathrm{Mean}\;\mathrm{grain}\;\mathrm{filling}\;\mathrm{rate}\;\mathrm{at}\;\mathrm{middle}\;\mathrm{grain}\;\mathrm{filling}\;\mathrm{period}\;(\mathrm{mg}\;{\mathrm{grain}}^{-1}\;d^{-1})\;=\frac{W_{2}-W_{1}}{t_{2}-t_{1}}$$
(19)$$\mathrm{Mean}\;\mathrm{grain}\;\mathrm{filling}\;\mathrm{rate}\;\mathrm{at}\;\mathrm{late}\;\mathrm{grain}\;\mathrm{filling}\;\mathrm{period}\;(\mathrm{mg}\;{\mathrm{grain}}^{-1}\;d^{-1})\;=\frac{W_{3}-W_{2}}{t_{3}-t_{2}}$$
(20)$$\mathrm{Contribution}\;\mathrm{rate}\;\mathrm{of}\;\mathrm{the}\;\mathrm{early}\;\mathrm{grain}\;\mathrm{filling}\;\mathrm{period}\;(\%)=\frac{W_{1}}{A_{1}}$$
(21)$$\mathrm{Contribution}\;\mathrm{rate}\;\mathrm{of}\;\mathrm{the}\;\mathrm{middle}\;\mathrm{grain}\;\mathrm{filling}\;\mathrm{period}\;(\%{)}_{\;}=\frac{W_{2}-W_{1}}{A_{2}-A_{1}}$$
(22)$$\mathrm{Contribution}\;\mathrm{rate}\;\mathrm{of}\;\mathrm{the}\;\mathrm{late}\;\mathrm{grain}\;\mathrm{filling}\;\mathrm{period}\;(\%{)}_{\;}=\frac{W_{3}-W_{2}}{A_{3}-A_{2}}$$

### 4.5. Microstructure Analysis

In 2020, mature rice seeds of different types of chalkiness were transversely cut using a knife and observed using a scanning electron microscope (Quanta 450; Thermo Fisher Scientific, Hillsboro, OR, USA) at an accelerating voltage in the range of 10–20 kV.

### 4.6. Appearance Quality

The CGR and CD of rice grains were measured using a rice appearance quality detector (JMWT 12, Beijing Dongfu Jiuheng Instrument Technology Co., Ltd., Beijing, China). In 2018 and 2019, chalkiness types were categorized as white-belly, white-core, white-back, and full-white chalkiness according to the location of the chalkiness, and grains with more than two types of chalkiness were described as having mixed-white chalkiness.

### 4.7. Statistical Analysis

The data were analyzed using the SPSS software v19.0 (IBM, Inc., Chicago, IL, USA). Significant differences between treatments were determined by performing Tukey’s tests, and statistical significance was set at *p* < 0.05. Further, figures were drawn using the GraphPad Prism software v8.0.1 (GraphPad Software, Inc., San Diego, CA, USA). The data shown in all tables and figures represent the mean of three replicates.

## 5. Conclusions

In this study, we observed that the CGR and CD were significantly negatively associated with the mean grain filling rates, maximum grain filling rates, grain weight at maximum grain filling rate, and mean grain filling rate of the rice grains but positively correlated with the time for reaching the maximum filling rate, effective grain filling period, and inflection point values. Furthermore, LL markedly decreased the activity of AGPP and those of SuS and SSS at the early grain filling stage and GBSS and SBE after 12 DAF, which led to insufficient starch synthesis in rice grains under LL. Therefore, LL significantly decreased the mean grain filling rates and maximum grain filling rates and increased the time for reaching the maximum filling rate and effective grain filling period, leading to slower grain filling and poorer grain fullness. Thus, both CGR and CD were significantly increased by LL owing to the increases in white-core, white-belly, white-back, mixed-white, and full-white chalkiness rates with more loosely arranged amyloplasts in grains. Furthermore, LL led to a greater increase in time for reaching the maximum filling rate, t_1_, t_2_, contribution rate of the middle grain filling period, and contribution rate of the late grain filling period, as well as a greater decrease in I, grain weight at maximum grain filling rate, mean grain filling rate during the early period, and contribution rate of the early grain filling period at the bottom panicle part than at the top of the spikelet. Therefore, a more substantial increase in CGR and CD was observed for the grains at the bottom panicle part under LL. Future research using LL stress should focus on reducing chalkiness by coordinating starch synthesis and enhancing the progress of grain filling of rice, particularly for grains at the bottom panicle part.

## Figures and Tables

**Figure 1 ijms-23-09153-f001:**
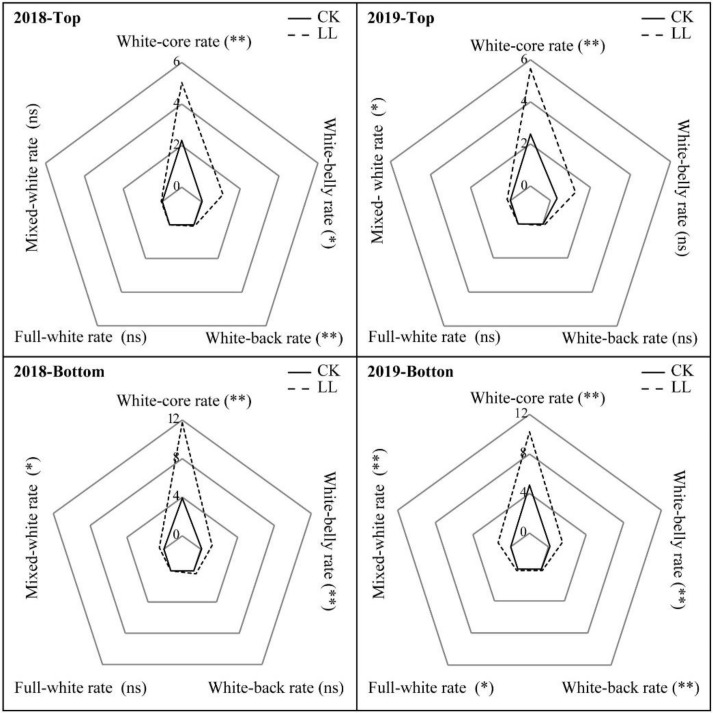
Variation in the rates of the chalkiness types of rice grains grown under LL stress in 2018 and 2019. CK, full sunlight control; LL, low light stress; * *p* < 0.05; ** *p* < 0.01; ns, no significance.

**Figure 2 ijms-23-09153-f002:**
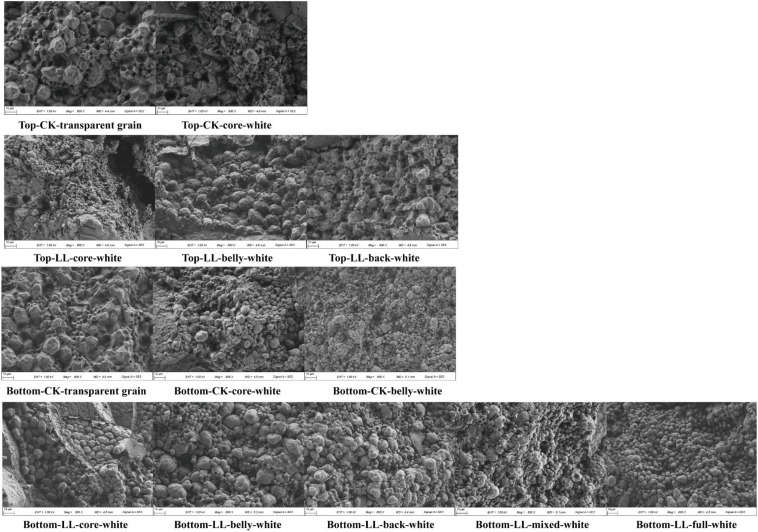
Changes in the microstructure of the grains in 2020. Top, top panicle part; bottom, bottom panicle part; CK, full sunlight control; LL, low light stress.

**Figure 3 ijms-23-09153-f003:**
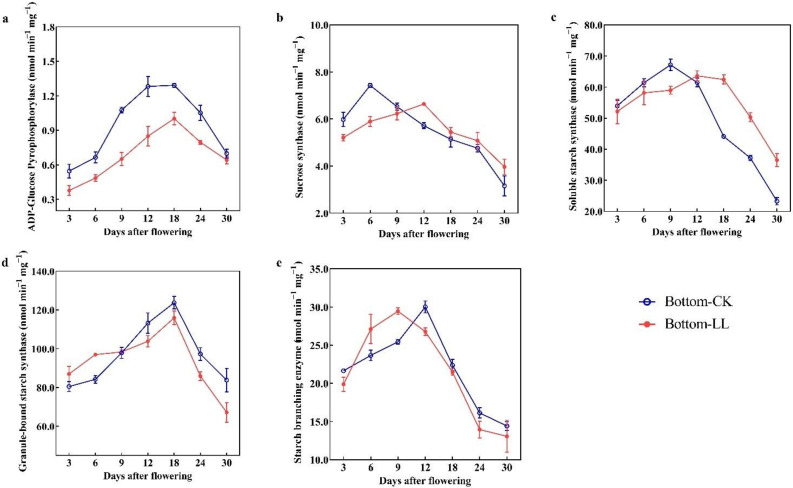
Activities of ADP-glucose pyrophosphorylase (**a**), sucrose synthase (**b**), soluble starch synthase (**c**), granule-bound starch synthase (**d**) and starch branching enzyme (**e**) in rice grains under LL in 2021. Bottom, bottom panicle part; CK, full sunlight (control); LL, low light stress.

**Figure 4 ijms-23-09153-f004:**
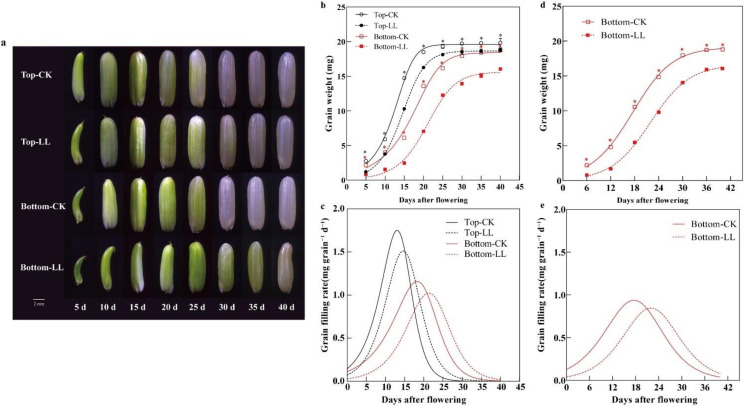
Growth and development of rice grain and grain filling under LL in 2020 (**a**–**c**) and 2021 (**d**,**e**). Top, top panicle part; bottom, bottom panicle part; CK, full sunlight control; LL, low light stress. * *p* < 0.05.

**Figure 5 ijms-23-09153-f005:**
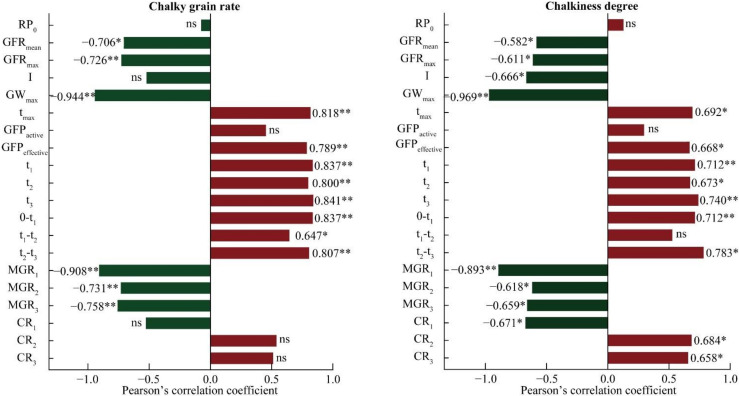
Relationships between chalkiness and rice grain filling characteristic (*n* = 12). RP_0_, relative initial growth potential; GFR_mean_, mean grain filling rate; GFR_max_, maximum grain filling rate; I, grain weight at maximum grain filling rate/maximum grain weight; GW_max_, grain weight at maximum grain filling rate; t_max_, time to reach the maximum filling rate; GFP_active_, active grain filling period; GFP_effective_, effective grain filling period; 0–t_1_, early period; t_1_–t_2_, middle period; t_2_–t_3_, late period; MGR_1_, mean grain filling rate during the early period; MGR_2_, mean grain filling rate during the middle period; MGR_3_, mean grain filling rate during the late period; CR_1_, contribution rate of the early period; CR_2_, contribution rate of the middle period; CR_3_, contribution rate of the late period; * *p* < 0.05; ** *p* < 0.01.

**Figure 6 ijms-23-09153-f006:**
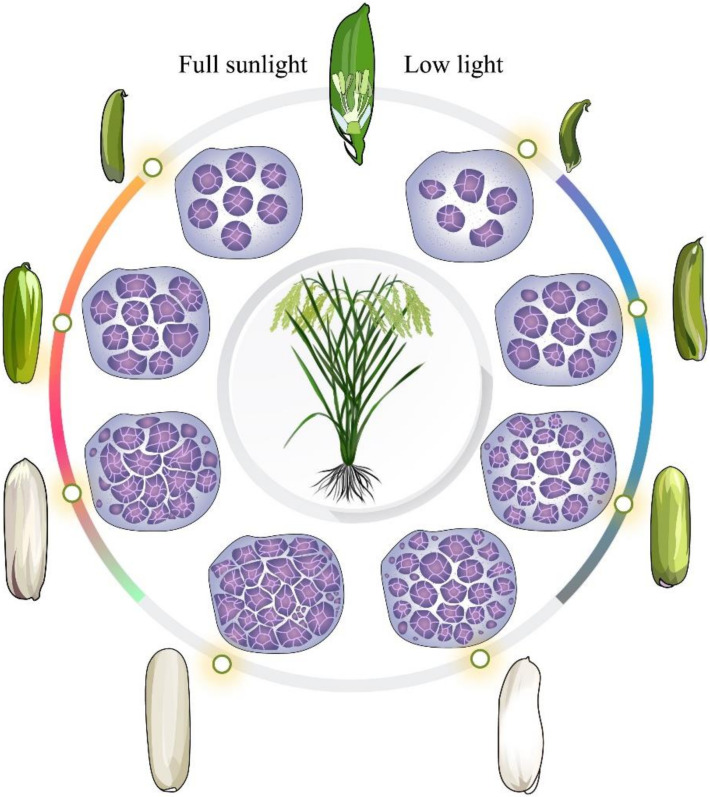
Pattern map depicting rice grain development.

**Figure 7 ijms-23-09153-f007:**
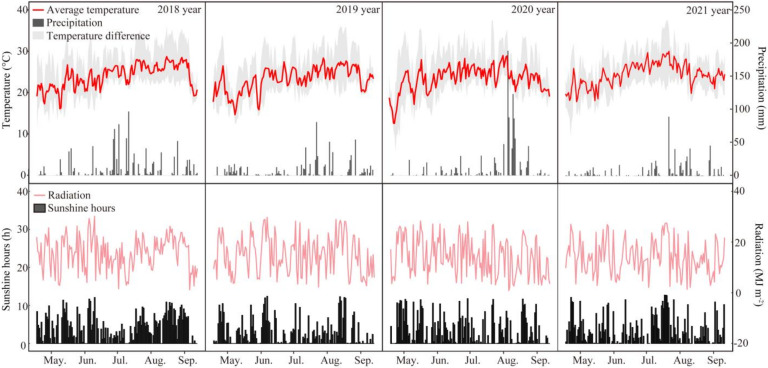
Meteorological data corresponding to the rice growth seasons.

**Table 1 ijms-23-09153-t001:** Effect of low light stress on chalky grain rate at different panicle parts.

Panicle Part	Light Treatment	Chalky Grain Rate (%)				Chalkiness Degree (%)			
		2018	2019	2020	2021	2018	2019	2020	2021
Top	CK	2.29 ± 0.020 ^b^	2.84 ± 0.003 ^b^	2.50 ± 0.011 ^b^	—	0.500 ± 0.006 ^a^	0.537 ± 0.002 ^b^	0.687 ± 0.002 ^b^	—
	LL	6.17 ± 0.011 ^a^	7.08 ± 0.005 ^a^	4.91 ± 0.001 ^a^	—	1.42 ± 0.005 ^a^	2.18 ± 0.005 ^a^	1.44 ± 0.001 ^a^	—
Bottom	CK	4.01 ± 0.016 ^b^	5.06 ± 0.013 ^b^	3.76 ± 0.001 ^b^	2.73 ± 0.005 ^b^	0.503 ± 0.007 ^b^	0.920 ± 0.004 ^b^	0.830 ± 0.001 ^b^	0.683 ± 0.002 ^b^
	LL	12.97 ± 0.035 ^a^	13.48 ± 0.006 ^a^	11.84 ± 0.004 ^a^	9.82 ± 0.004 ^a^	2.13 ± 0.005 ^a^	2.43 ± 0.004 ^a^	2.23 ± 0.001 ^a^	1.74 ± 0.003 ^a^
F value	Panicle part (P)	11.03 *	96.70 **	145.41 **	—	1.23 ns	2.04 ns	48.85 **	—
	Light treatment (L)	24.99 **	208.49 **	239.21 **	339.29 **	15.41 **	50.42 **	261.62 **	29.58 **
	P × L	3.92 ns	22.71 **	69.83 **	—	1.21 ns	0.091 ns	23.45 **	—

Lowercase letters represent significant difference between CK and LL at the same panicle part (*p* < 0.05). CK, full sunlight (control); LL, low light stress; *, *p* < 0.05; **, *p* < 0.01; ns, no significance.

**Table 2 ijms-23-09153-t002:** Stimulation equations of grain filling of different panicle parts under low light stress.

Year	Panicle Part	Light Treatment	*R* ^2^	Stimulation Equations
2020	Top	CK	0.9966	W_G_ = 19.64(1 + 668.10e^−0.4503t^)^−1/1.88^
		LL	0.9992	W_G_ = 18.68(1 + 204.90e^−0.0.3484t^)^−1/1.25^
	Bottom	CK	0.9923	W_G_ = 18.45(1 + 930.00e^−0.3334t^)^−1/2.10^
		LL	0.9955	W_G_ = 15.63(1 + 679.00e^−0.2916t^)^−1/1.37^
2021	Bottom	CK	0.9985	W_G_ = 19.11(1 + 44.00e^−^^0.2063t^)^−^^1/1.17^
		LL	0.9986	W_G_ = 16.62(1 + 97.16e^−^^0.2061t^)^−^^1/1.04^

Note: CK, full sunlight (control); LL, low light; W_G_, the dry weight of the grain; t, the time after flowering.

**Table 3 ijms-23-09153-t003:** Grain filling characteristics at different panicle parts.

Year	Panicle Part	Light Treatment	RP_0_	GFR_mean_ (mg·Grain^−1^·d^−1^)	GFR_max_ (mg·Grain^−1^·d^−1^)	I (%)	GW_max_ (mg)	t_max_ (Day)	GFP_active_ (Day)	GFP_effective_ (Day)
2020	Top	CK	0.240 ± 0.020 ^a^	1.14 ± 0.003 ^a^	1.75 ± 0.017 ^a^	56.93 ± 0.021 ^a^	11.18 ± 0.307 ^a^	13.05 ± 0.285 ^b^	17.21 ± 0.175 ^b^	24.64 ± 0.419 ^b^
		LL	0.283 ± 0.042 ^a^	1.00 ± 0.030 ^b^	1.51 ± 0.041 ^b^	52.22 ± 0.023 ^a^	9.75 ± 0.339 ^b^	14.63 ± 0.283 ^a^	18.65 ± 0.642 ^a^	28.43 ± 0.576 ^a^
	Bottom	CK	0.160 ± 0.000 ^b^	0.751 ± 0.012 ^a^	1.16 ± 0.022 ^a^	58.36 ± 0.007 ^a^	10.77 ± 0.141 ^a^	18.28 ± 0.095 ^b^	24.56 ± 0.358 ^a^	34.27 ± 0.453 ^b^
		LL	0.210 ± 0.010 ^a^	0.677 ± 0.017 ^b^	1.03 ± 0.030 ^b^	53.30 ± 0.014 ^b^	8.33 ± 0.105 ^b^	21.29 ± 0.111 ^a^	23.12 ± 0.860 ^a^	38.12 ± 0.941 ^a^
F-value	Panicle part (P)		31.58 **	1122.78 **	1060.94 **	1.62 ns	42.14 **	2322.37 **	320.09 **	700.80 **
	Light treatment (L)		11.70 **	100.28 **	126.49 **	24.63 **	186.85 **	346.53 **	0.000 ns	109.67 **
	P × L		0.060 ns	9.19 *	9.97 *	0.031 ns	12.54 **	33.10 **	18.97 **	0.008 ns
2021	Bottom	CK	0.193 ± 0.006 ^a^	0.812 ± 0.005 ^a^	1.24 ± 0.010 ^a^	0.550 ± 0.007 ^a^	10.48 ± 0.099 ^a^	21.23 ± 0.199 ^b^	23.48 ± 0.202 ^b^	37.77 ± 0.170 ^b^
		LL	0.197 ± 0.006 ^a^	0.649 ± 0.013 ^b^	0.982 ± 0.023 ^b^	0.529 ± 0.011 ^b^	8.73 ± 0.180 ^b^	23.86 ± 0.130 ^a^	25.45 ± 0.512 ^a^	42.53 ± 0.619 ^a^
F-value	Light treatment (L)		1.20 ns	402.96 **	301.21 **	8.44 *	218.49 **	367.17 **	38.50 **	165.09 **

Lowercase letters represent significant difference between CK and LL at the same panicle part (*p* < 0.05). RP_0_, relative initial growth potential; GFR_mean_, mean grain filling rate; GFR_max_, maximum grain filling rate; I, grain weight at maximum grain filling rate/maximum grain weight; GW_max_, grain weight at maximum grain filling rate; t_max_, time for reaching the maximum filling rate; GFP_active_, active grain filling period; GFP_effective_, effective grain filling period; CK, full sunlight (control); LL, low light stress; *, *p* < 0.05; **, *p* < 0.01; ns, no significance.

**Table 4 ijms-23-09153-t004:** Grain filling characteristics at different stages and panicle parts.

Year	Panicle Part	Light Treatment	t_1_	t_2_	t_3_	0–t_1_	t_1_–t_2_	t_2_–t_3_	MGR_1_	MGR_2_	MGR_3_	CR_1_	CR_2_	CR_3_
2020	Top	CK	9.63 ± 0.434 ^b^	16.46 ± 0.147 ^b^	23.25 ± 0.653 ^b^	9.63 ± 0.434 ^b^	6.83 ± 0.308 ^b^	6.80 ± 0.769 ^b^	0.606 ± 0.022 ^a^	1.55 ± 0.017 ^a^	0.450 ± 0.011 ^a^	29.72% ± 0.026 ^a^	53.76% ± 0.014 ^a^	15.52% ± 0.013 ^a^
		LL	10.66 ± 0.337 ^a^	18.61 ± 0.307 ^a^	27.85 ± 0.910 ^a^	10.66 ± 0.337 ^a^	7.95 ± 0.309 ^a^	9.24 ± 0.940 ^a^	0.416 ± 0.031 ^b^	1.33 ± 0.035 ^b^	0.377 ± 0.009 ^b^	23.81% ± 0.027 ^a^	56.58% ± 0.012 ^a^	18.61% ± 0.016 ^a^
	Bottom	CK	13.52 ± 0.198 ^b^	23.04 ± 0.104 ^b^	32.03 ± 0.516 ^b^	13.52 ± 0.198 ^b^	9.51 ± 0.252 ^a^	8.99 ± 0.440 ^a^	0.430 ± 0.007 ^a^	1.02 ± 0.020 ^a^	0.300 ± 0.007 ^a^	31.56% ± 0.009 ^a^	52.82% ± 0.005 ^b^	14.63% ± 0.004 ^b^
		LL	16.43 ± 0.387 ^a^	26.15 ± 0.167 ^a^	37.06 ± 1.290 ^a^	16.43 ± 0.387 ^a^	9.72 ± 0.554 ^a^	10.91 ± 1.124 ^a^	0.239 ± 0.008 ^b^	0.902 ± 0.027 ^b^	0.257 ± 0.009 ^b^	25.12% ± 0.017 ^b^	56.02% ± 0.008 ^a^	17.87% ± 0.010 ^a^
F-value	Panicle part (*p*)		570.41 **	3878.32 **	304.50 **	570.41 **	106.20 **	643.94 **	241.13 **	1019.97 **	1.58 ns	1.64 ns	1.66 ns	15.25 **
	Light treatment (L)		94.29 **	537.84 **	87.09 **	94.29 **	9.45 *	120.41 **	281.02 **	130.20 **	23.62 **	25.30 **	26.74 **	19.43 **
	P × L		21.44 **	18.15 **	0.183 ns	21.44 **	4.41 ns	7.77 *	0.0120 ns	9.99 *	0.013	0.047 ns	0.110 ns	0.283 ns
2021	Bottom	CK	16.41 ± 0.289 ^b^	26.04 ± 0.109 ^b^	36.25 ± 0.299 ^b^	16.41 ± 0.289 ^b^	9.63 ± 0.180 ^b^	10.21 ± 0.408 ^b^	0.316 ± 0.003 ^a^	1.09 ± 0.010 ^a^	0.313 ± 0.004 ^a^	27.18 ± 0.008 ^a^	55.06 ± 0.004 ^b^	16.76 ± 0.004 ^b^
		LL	18.47 ± 0.303 ^a^	29.24 ± 0.109 ^a^	41.48 ± 0.920 ^a^	18.47 ± 0.303 ^a^	10.77 ± 0.374 ^a^	12.23 ± 0.841 ^a^	0.220 ± 0.008 ^b^	0.863 ± 0.021 ^b^	0.245 ± 0.007 ^b^	24.58 ± 0.013 ^b^	56.27 ± 0.006 ^a^	18.15 ± 0.007 ^a^
F-value	Light treatment (L)		72.73 **	1286.97 **	87.44 **	72.73 **	22.49 **	14.09 *	356.83 **	280.21 **	196.03 **	8.49 *	8.59 *	8.39 *

Lowercase letters represent significant difference between CK and LL at the same panicle part (*p* < 0.05). 0–t_1_, early period; t_1_–t_2_, middle period; t_2_–t_3_, late period; MGR_1_, mean grain filling rate at early period; MGR_2_, mean grain filling rate at middle period; MGR_3_, mean grain filling rate at late period; CR_1_, contribution rate of the early period; CR_2_, contribution rate of the middle period; CR_3_, contribution rate of the late period; CK, full sunlight (control); LL, low light stress; *, *p* < 0.05; **, *p* < 0.01; ns, no significance.

**Table 5 ijms-23-09153-t005:** Soil properties of experimental sites between 2018 and 2021.

Year	Organic Matter (g kg^−1^)	Total N (g kg^−1^)	Total P (g kg^−1^)	Total K (g kg^−1^)	Alkaline Hydrolysable N (mg kg^−1^)	Olsen P (mg kg^−1^)	Exchangeable K (mg kg^−1^)
2018	28.50	1.50	0.94	17.50	137.20	50.82	143.10
2019	29.92	1.47	1.04	16.43	72.65	43.83	164.10
2020	27.90	1.63	0.99	18.10	89.76	54.68	158.40
2021	27.34	1.98	0.95	17.86	21.07	53.91	126.81

## Data Availability

Not applicable.

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
