# Peer review of "Low Light Stress Increases Chalkiness by Disturbing Starch Synthesis and Grain Filling of Rice"

_ijms, 2022, doi:10.3390/ijms23169153_

Round 1
Reviewer 1 Report
Chalkiness is an important quality character that affects the commodity quality of rice. The study in this manuscript on the effect of low light stress on chalkiness of rice is of positive significance to the cultivation of high quality rice.The authors studied the relationship between the filling characteristics and chaliness peformance in one rice variety at the flowering stage for four consecutive years by artificial shanding treatment as the low light environment. The amount of work is large and the data is rich, which is of positive signigicance for the development of high quality cultivation measures of rice varieties.The manuscript also has the following problems to be further enriched and improved:
1. The anthors only selected one indica rice variety, Huanghuazhan, as the test manterial, is it universal to other indica rice varieties or japonica rice varitieties?
2. Low light stress is the important feature of the manuscript. It is very important to determine the standard of low light stress. I noticed that the authors described as followed: a layer of white cotton yarn screens was hung at approximately 0.5 m above the rice plants from flowering to maturity following the method described by Deng et[5,8]'. The two citations appear to have come from the same laboratory as the manuscript. As far as I know, other studies have shown that appropriate shading treatment is beneficial to rice varieties ,especially the improvement of appearance and processing quality, which seems to be different from the results of this manuscript. According to my speclation. the experiment conducted in Sichuan province (30o43N,103o52N, Wenjiang, Sichuan, China),may not have high light intensity at the late growth stage of rice. Therefore, artificial shading treament of 50% might cause low light stress,but for the middle and lower reaches of The Yangtze River in China,it's probably not the case. It is very improtant to indicate the intensity of light at the experiment. I noticed that the authors indeed listed the related information in the "materials and methods"(page 16) in Figure 7 as radiation(mm). It seemed tha the "mm" is not the internation units of radiation. It is a bit confusing to the reader. It is a bit difficult to determine the level of light condition,even to comare other rice-growing areas. Therefore, I strongly recomment that the anthors replace the data for radiation as the more general international units in this manuscript.
3 Chalkiness is a quantitative trait that is controlled by multiple genes, such as Chalk 5, OsPK2, qPCG1 and qACE9. If the manuscript could add some research on the genes mentioned above associated with chalkness, such as the dynamics of their genes during the grain filling stage. It would be more tageted than the paralled correlation between chalkiness phenotype and filling propeties. This part would be helpful to increase the innovation of this manuscript.
Reviewer 2 Report
General comments:
Overall this manuscript tried to address an important quality deficiency in rice production. Experiments were designed in four consecutive years, however, not all data were collected in all years (e.g. enzymatic activity analysis, and microstructure analysis only performed in one year). There are lots of abbreviations which made it hard to follow. Results section need a revision to clarify some confusing points. Discussion needs improvement. Reference section needs major revision.
Specific comments:
Abstract:
Avoid using abbreviations in abstract, especially those that were not used afterward.
Introduction:
lots of abbreviations made it hard to follow. Line72. I guess “SS” should be “SSS”
Results:
Table 1 title is “Effect of low light stress on chalky rice rate at different panicle parts.” The words “chalky rice rate” should be revised, it is not consistent with the text.
Lines 96-99, is a repetition of what already presented on lines 91-94.
Lines110-111, how these called “improved”?? all mentioned traits were increased (which are undesirable)
Figure 2. the quality of photos are poor, and the photo magnification scale is unreadable.
Line 134. “AGPase” was introduced here for the first time without full name explanation.
Figure 3. why only bottom panicle part results were presented in this figure? While in the text (lines 132-142), there is no clear explanation of Top and Bottom panicle parts comparison, and why only bottom part was sampled. Why this test only performed in the last year of their study (2021)?
Line133-134. “Light conditions markedly influenced the activities of these enzymes”, not for all dates (e.g. at day 30, the AGPase of CK and LL seems the same). Revise the sentence.
Table 2. A, B, C, and D values presented in the Table were presented again on the same Table in the “stimulation equations”; they can be deleted from the Table.
Table 3. the title mentioned “parameters”, while these referred to “characteristics” in the text. Also, in the Table, “E” defined as “Effective grain filling period”, while in the M&M section (Line 395) it is abbreviated as “GFPeffective”. Please be consistent.
Figure 4. Use “ns” on the values that show no significant relationships.
Discussion:
Focuses more on other people works, rather than explaining/discussing their results and their relevance to the previous findings; discussion needs improvement.
Materials and Methods:
Line 341. what time (month and day) plants were transferred to plots in each year?
Line 347. Specify the type of pesticides used
Table 5. Are these measurements performed once, in each year? Or they are average of several measurements? What was the purpose of this analysis?
References:
The format used, does not match with the Journal guidelines, examples:
Author names should be in this format: Author 1, A.B.; Author 2, C.D.
Year should not be in the parenthesis
Round 2
Reviewer 1 Report
The anthor carefully revised the manuscript and explained my comments.
The changes of grain-filling related genes were not involved in the original experimental design, it is a pity. It is understandable that it can nont be supplemented temporarily because it is a four-year experiment. I hope it can ben seen in the author's future work.
I know that Huanghuazan is a conventional indica rice variety with a large popularization area and good appearance in South China. The anthor's current research is ver meaningful.What I mean in the original comment by that is that choosing anthor poor appearance rice variety may be more valuable in good cultivation.